# Waterborne and Dietary Bioaccumulation of Organophosphate Esters in Zooplankton *Daphnia magna*

**DOI:** 10.3390/ijerph19159382

**Published:** 2022-07-31

**Authors:** Wenxin Liu, Hong Zhang, Jiaqi Ding, Wanyu He, Lin Zhu, Jianfeng Feng

**Affiliations:** Key Laboratory of Pollution Process and Environmental Criteria of Ministry of Education and Tianjin Key Laboratory of Environmental Remediation and Pollution Control, College of Environmental Science and Engineering, Nankai University, Tianjin 300071, China; wenxinliu1621@163.com (W.L.); lookatzhanghong@163.com (H.Z.); dingjiaqi9823@163.com (J.D.); hewanyu123456@163.com (W.H.)

**Keywords:** organophosphate esters, bioaccumulation, dietary exposure, waterborne exposure, toxicokinetics

## Abstract

Organophosphate esters (OPEs) are widely used as an additive in flame retardants, plasticizers, lubricants, consumer chemicals, and foaming agents. They can accumulate in aquatic organisms from water (waterborne exposure) and food (dietary exposure). However, the bioaccumulation characteristics and relative importance of different exposure routes to the bioaccumulation of OPEs are relatively poorly understood. In this study, *Daphnia magna* were exposed to fo typical OPEs (tris(2-chloroethyl) phosphate (TCEP), tris(1,3-dichloro-2-propyl) phosphate (TDCIPP), tris(2-butoxyethyl) phosphate (TBOEP), and triphenyl phosphate (TPHP)), and their toxicokinetics under waterborne and dietary exposure routes were analyzed. For the waterborne exposure route, the bioconcentration factors (BCFs) increased in the order of TBOEP, TCEP, TDCPP, and TPHP, which were consistent with their uptake rate constants. TPHP might have the most substantial accumulation potential while TBOEP may have the smallest potential. In dietary exposure, the depuration rate constants of four OPEs were different from those in the waterborne experiment, which may indicate other depuration mechanisms in two exposure routes. The biomagnification factors (BMFs) of fur OPEs were all below 1, suggesting trophic dilution in the transfer of four OPEs from *Scenedesmus obliquus* to *D. magna*. Except for TBOEP, the contributions of dietary exposure were generally lower than waterborne exposure in *D. magna* under two exposure concentrations. This study provides information on the bioaccumulation and contribution of OPEs in *D. magna* via different exposure routes and highlights the importance of considering different exposure routes in assessing the risk of OPEs.

## 1. Introduction

As a new type of phosphorus flame retardant, organophosphate esters (OPEs) have the characteristics of low smoke, low toxicity, and low halogen compared with traditional brominated flame retardants. Therefore, they are widely used in flame retardants, plasticizers, lubricants, consumer chemicals, and foaming agents as additives. However, since the addition method is often physical addition rather than chemical bonding, they easily diffuse into the surrounding environment [1]. OPEs have been widely detected both in freshwater and marine environment and their food webs, such as Taihu Lake [2,3,4], the freshwater environment in South China [5], Yangtze River Basin [6], Laizhou bay [7,8], Southern European waters [9], surface and tap waters in New York State [10], inland and coastal waters in northern Greece [11], and Antarctic [12]. In addition to being detected in the natural environment, many OPEs such as tris(2-chloroethyl) phosphate (TCEP), tris-2-chloroisopropyl phosphate (TCPP), tris(1,3-dichloro-2-propyl) phosphate (TDCIPP), tris(2-butoxyethyl) phosphate (TBOEP), tri-n-butyl phosphate (TNBP), and triphenyl phosphate (TPHP) have been proven to possess neurotoxicity [13,14], endocrine disruption [15], viscera toxicity [16], genetic toxicity [1], or mutagenicity [17].

For aquatic organisms, the routes that pollutants enter the body mainly include the bioaccumulation of pollutants by filtering the surrounding water (waterborne route) and the absorption and assimilation of lower-trophic organisms (dietary route). Since these pollutants enter the body through different pathways, their uptake and depuration processes and their effects may be different. For example, Guo et al. found that in fish (*Mugilogobius chulae*), the uptake and contribution of dietary and waterborne cadmium (Cd), was positively correlated with the Cd concentration in the chyme [18]. The different relative importance of hematite nanoparticles (NPs) in tissues via waterborne and dietary exposure was reported in zebrafish [19], and waterborne exposure caused higher mortality, while the lower neonate production of Daphnia was observed in dietary exposure to Copper oxide nanoparticles [20]. De Wit et al. also found that when waterborne exposure reached a steady state, dietary transfer controlled the bioaccumulation of MeHg [21]. It has been reported that some OPEs such as TCPP, TBOEP, TCEP, TDCPP, and TPHP have biomagnification effects [7,12], which means they can be transferred to organisms at a higher trophic level and may finally have a negative impact on human health. However, the bioaccumulation process and level via waterborne and dietary exposure may also be different, and the relative importance of the two exposure pathways in the organism is unclear. Considering the potential biomagnification effects of OPEs, it is necessary to study the kinetics and contributions of OPEs via waterborne and dietary exposure in aquatic organisms, which may help to understand which exposure pathway of OPEs is more critical in organisms at higher trophic levels.

In the natural aquatic food webs, dietary exposure is mainly manifested in the form of the food chain. Phytoplankton and zooplankton are the leading producers and critical consumers of aquatic ecosystems, respectively. Zooplankton plays a vital ecological role in the aquatic food web, linking essential resources to higher-level consumers [22]. *Scenedesmus obliquus* is an important producer at the first level of the trophic chain in the food web [23]. It is the most widely distributed freshwater microalgae species [24]. *Daphnia magna* is filter-feeding zooplankton. As a primary consumer in water bodies, *D. magna* feeds on algae and debris in water bodies. Due to their fast reproduction, easy cultivation, and short life cycle, they have become model zooplankton species for aquatic ecotoxicology research [25]. The effects of TDCPP [26], TCEP [27], TBOEP [28], and TPHP [29] on *D. magna* have been investigated. Except for TCEP, the other three OPEs had negative impacts on the growth, reproduction, and survival of *D. magna*. Furthermore, TCEP, TDCPP, TBOEP, and TPHP might have potential biomagnification effects according to field research [7,12]. Therefore, in this study, we studied and compared the transfer of TCEP, TDCPP, TBOEP, and TPHP from a solution and *Scenedesmus obliquus* to *Daphnia magna*. The primary purpose is to (1) explore the uptake and depuration kinetics of OPEs in *D. magna* via waterborne exposure and dietary exposure with the *S. obliquus*–*D. magna* food chain and (2) explore the relative importance of the two exposure routes to the accumulation of four OPEs in *D. magna*.

## 2. Materials and Methods

### 2.1. Chemicals and Organisms

TCEP, TDCPP, TBOEP, TPHP, and surrogate standards (d12-TCEP, d27-TNBP, and d15-TPHP) were purchased from Toronto Research Chemicals (Toronto, Canada). High-performance liquid chromatography (HPLC)-grade dimethyl sulfoxide (DMSO) and dichloromethane (DCM) were purchased from Bohua Chemical Reagent (Tianjin) Co., Ltd. (Tianjin, China). HPLC-grade acetonitrile (ACN) came from Fisher Scientific Co. (Fair Lawn, NJ, USA). HPLC-grade formic acid was purchased from Aladdin Biochemical Technology Co., Ltd. (Shanghai, China). HPLC-grade hexane, methanol, GCB/NH_2_ cartridges (200 mg/200 mg, 3 mL), and Envi-18 cartridges (500 mg, 6 mL) were purchased from Anpel Laboratory Technologies Inc. (Shanghai, China).

*Scenedesmus obliquus* was purchased from the National Freshwater Algae Seed Bank, Wuhan Institute of Hydrobiology, Chinese Academy of Sciences, and was incubated in flasks with BG11 artifact freshwater. *Daphnia magna* was obtained from the College of Environmental Science and Technology, Nankai University, and was maintained in 2 L beakers with 1.5 L of reconstituted water (KCl, 1.2 mg/L; CaCl_2_·2H_2_O, 58.6 mg/L; NaHCO_3_, 13.0 mg/L; and MgSO_4_·2H_2_O, 24.5 mg/L). Both *S. obliquus* and *D. magna* were cultured in a greenhouse at 25 ± 1 °C under a 16:8 h light/dark photoperiod. *D. magna* were fed *S. obliquus* daily, and 70% of the reconstituted water was refreshed twice a week.

### 2.2. S. obliquus Exposure to OPEs

Algae cells were prepared to an initial density of 5 × 10^6^ cells/mL in BG11 (300 mL, n = 3), then spiked with 0.01% OPEs stock solutions in DMSO to obtain concentrations of 20 and 100 μg/L and cultured in conditions as described earlier. The solvent control was prepared with an equal volume of DMSO. All experimental devices were cultured in a greenhouse for 96 h, which were shaken 3–4 times daily. Each experiment group was constructed in triplicate. Samples were collected at 0, 24, 48, 72, and 96 h. At each time point, 15 mL of algae liquid was collected and filtered by a GF/F membrane (pore size: 0.7 μm, diameter: 47 mm) to separate *S. obliquus* and the exposure solution. The exposure solution was transferred into glass tubes. The GF/F membrane with *S. obliquus* was rinsed three times using ultrapure water and stored at −20 °C for analysis. Test results are expressed as dry weight (dw).

### 2.3. D. magna Waterborne Exposure to OPEs

Waterborne exposure experiments consisted of a 24 h uptake period followed by a 24 h depuration period and were conducted in a solution of 20 and 100 μg/L OPEs. Each experiment group was constructed in triplicate. In the uptake period, after 24 h without feeding, 20 daphnia individuals (approximately 25 days old) and 400 mL of reconstituted water with OPEs were added to 500 mL beakers. All experimental devices were cultured in a greenhouse. During the experiment, the *D. magna* were not fed. Sampling was performed at 0, 3, 6, 12, and 24 h. Then, the remaining *D. magna* was transferred into clean reconstituted water for the depuration experiment. Samples were collected at 3, 6, 12, and 24 h. At each time point, 20 *D. magna* individuals were taken in triplicate, and then they were rinsed three times using ultrapure water and stored at −20 °C for analysis. The exposure solutions were also sampled at each time point to determine the OPEs concentration. Test results are expressed as dry weight (dw).

### 2.4. D. magna Dietary Exposure to OPEs

Algae with a density of 5 × 10^6^ cells/mL were exposed to 0, 20, and 100 μg/L OPEs in the BG11 culture medium for 24 h as described above for the algal uptake experiment. After 24 h without feeding, 20 daphnia individuals (approximately 25 days old) and 30 mL of clean reconstituted water were added to 50 mL glass tubes. We replaced the reconstituted water every 3 h and fed new algae pre-exposed for 24 h to ensure that the OPE in the food was not released into the water. According to the filtration rate (1.5 mL h^−1^ Ind.^−1^) calculated by us, the filtration time of the 30 mL solution was 1 h for 20 daphnia individuals, so they could absorb the algae in the solution to the maximum extent during 3 h. The OPEs-contaminated algae were obtained through centrifugation at 6500× *g* for 5 min at 4 °C and then washed once with ultrapure water. *D. magna* were fed at a density of 4 × 10^6^ cells/mL for 12 h. Each experiment group was constructed in triplicate. *S. obliquus*, *D. magna*, and solution samples were collected at 0, 1, 3, 6, 9, and 12 h as described above. Then they were stored at −20 °C for analysis. Test results of *S. obliquus* and *D. magna* are expressed as dry weight (dw).

### 2.5. Sample Preparation

*S. obliquus* and *D. magna* samples were freeze-dried at −50 °C for 24 h, weighed, cut off, and then transferred into 10 mL glass tubes for OPEs determination. The extraction process was performed according to previous studies [3,30] with a minor modification. In brief, after the sample was homogenized, 5 ng of an internal standard surrogate was added to the sample. Then, ACN was added for extraction. The supernatant was dried with nitrogen and purified with GCB/NH_2_ cartridges. More details are attached in the Appendix A.

### 2.6. Chemical Analysis and Quality Assurance and Control

The analysis of OPEs was performed according to previous studies [3,30]. The compounds were analyzed using UPLC−MS/MS (Xevo TQ-S, Waters, Milford, MA, USA) with a BEH C18 column (2.1 mm × 50 mm, 1.7 μm, Waters) coupled with a VanGuard precolumn (C18 column, 2.1 mm × 5 mm, 1.7 μm). More details are attached in the Appendix A. Milli-Q water (A) and ACN (B), both containing 0.1% formic acid, were used as binary eluent.

In order to avoid contamination, all glass tubes used in the experiment were soaked in an H_2_SO_4_-K_2_Cr_2_O_7_ solution for more than 24 h before use. Then they were rinsed three times with methanol and Mili-Q water. Before use, the GF/F filter membrane was fired in a muffle furnace at 400 °C for 6 h. The recoveries in *S. obliquus*, *D. magna*, and the solution (Appendix A) were measured by being spiked with 20 ng of TCEP, TDCPP, TBOEP, and TPHP, ranging from 73.40 ± 2.06% to 113.34 ± 3.02%. Values three times the signal-to-noise ratio were defined as the method detection limits (MDLs). Values ten times the signal-to-noise ratio were defined as the method quantification limits (MQLs) (Appendix A). More details are attached in the Appendix A.

### 2.7. Toxicokinetics Model

For the uptake and depuration kinetics, we used a first-order one-compartment toxicokinetics model to estimate the uptake and depuration constant rates in waterborne and dietary exposure for *D. magna*. The equations were constructed in the R program (R Core Team (2020), http://www.R-project.org/ (accessed on 10 May 2022)) with the packages “deSolve” and “simecol” according to our previous studies [31,32].

As for *D. magna*, in the waterborne experiment, Equation (1) was fitted to the observed concentration to estimate the uptake and depuration rate constant simultaneously, which referred to a previous study with a slight modification [33]:(1)$$dC_{t1}/dt=\{\begin{array}{cc} k_{u1}\cdot C_{w}-k_{d1}\cdot C_{t1}, & t\leq 24\\ -k_{d1}\cdot C_{t1}, & t>24 \end{array}$$
where *t* is the time (h), *C_t_*_1_ is the internal concentration of OPEs in *D. magna* (ng/g) via waterborne exposure, *C_w_* is the concentration of OPEs in the exposure solution (μg/L), and *k_u_*_1_ and *k_d_*_1_ are the uptake rate constant (L kg^−1^ h^−1^) and the depuration rate constant (h^−1^), respectively.

The bioconcentration factors (BCF, L kg^−1^) in daphnia were calculated from the kinetic parameters following Equation (2):(2)$$\mathrm{BCF}=k_{u1}/k_{d1}$$

In the dietary experiment, Equation (3) was fitted to the observed concentration to estimate the uptake and depuration rate constant simultaneously, which referred to a previous study with a slight modification [33]:(3)$$dC_{t2}/dt=k_{u2}\cdot C_{so}-k_{d2}\cdot C_{t2}$$
where *t* is the time (h), *C_t_*_2_ is the internal concentration of OPEs in daphnia (ng/g) via dietary exposure, *C_so_* is the average concentration of OPEs in *S. obliquus* (ng/g) every three hours, and *k_u_*_2_ and *k_d_*_2_ are the uptake rate constant (g g^−1^ h^−1^) and the depuration rate constant (h^−1^), respectively.

The biomagnification factors (BMF) in daphnia were calculated from the kinetic parameters following Equation (4), which referred to a previous study [23]:(4)$$\mathrm{BMF}=k_{u2}/k_{d2}$$

In our study, we assumed that the internal concentration of algae at 24 h was steady under their exposure concentration, and that the process of accumulation via waterborne and dietary exposure does not interfere with each other. Then we could calculate the contribution of OPEs in daphnia via dietary exposure (F) following Equation (5) according to a previous study [34]:*F = C_t_*_2_/(*C_t_*_1_ + *C_t_*_2_) (5)
where *C_t_*_1_ is the internal concentration of OPEs in daphnia (ng/g) via waterborne exposure and *C_t_*_2_ is the internal concentration of OPEs in daphnia (ng/g) via dietary exposure.

### 2.8. Statistical Analysis

Each experiment group was constructed in triplicate in both waterborne and dietary exposure experiments. The error bar represents the standard error.

## 3. Results

### 3.1. Accumulation of OPEs in D. magna via Waterborne Route

During the 24 h uptake period, the internal concentration of four OPEs in *D. magna* increased rapidly within 6 h (Figure 1). The exposure concentration had an impact on the level of OPEs in *D. magna*, which is consistent with fish [35]. Among the four OPEs, TPHP has the highest uptake rate constant (*k_u_*_1_) (Table 1), reaching 21.15 and 138.80 L kg^−1^ h^−1^, and TBOEP has the lowest one, obtaining 0.17 and 0.57 L kg^−1^ h^−1^.

During the 24 h depuration period, the internal concentration of OPEs decreased rapidly in the first 6 h (Figure 1). TPHP has the slowest depuration rate and the longest half-life (Table 1), likely because it has the strongest hydrophobicity among the four OPEs and is more difficult to transfer to the external solution. The depuration route mainly includes transfer to the outside medium or biodegradation [30]. Previous studies have reported that the biodegradation process of OPEs and organophosphate tri-esters would be transformed into diesters and hydroxylated triesters, and chlorinated OPEs are relatively difficult to biodegrade [36]. TCEP was reported to be less accumulative and resistant to metabolism than TBOEP in salmon [37]. The above results suggested that TCEP and TDCPP should be mainly transferred to the culture medium, while biotransformation might contribute more to the removal process of non-chlorinated OPEs, which may be the reason for their different depuration rates in *D. magna*. In future research, further experiments should be constructed to determine the proportion and difficulty of the biotransformation of these four OPEs in *D. magna*.

The BCFs of TCEP, TDCPP, TBOEP, and TPHP in *D. magna* (Table 1) were higher than in *Danio rerio* [38], *Cyprinus carpion* [35,39], and *Mytilus galloprovincialis* [40], indicating the accumulation of OPEs in zooplankton may be stronger than in invertebrate and fish. A similar phenomenon was also found in a lake food web [2]. The total OPEs concentration in biota generally decreased in the order of plankton, invertebrates, and fish. Wang et al. illustrated BCFs of TCEP and TDCIPP were low among seven OPEs with a range of 0.5 to 66 and little bioaccumulation potential of TBOEP (BCF = 17.3 L kg^−1^) and strong bioaccumulation potential of TPHP (BCF ranged from 45 to 224 L kg^−1^) in zebrafish [38]. Furthermore, TCEP and TBOEP were relatively less bioaccumulative in carp, with BCF ranging from 1.0 ± 0.1 to 14.8 ± 0.2 [39]. Meanwhile, Mata et al. found a lower accumulation of TBOEP (BCF = 207 L kg^−1^) and a higher accumulation of TPHP (BCF = 3685 L kg^−1^) in mussel [40]. These results were all analogous to our study, indicating that the similarity of the accumulative capacity of OPEs among different species may be determined by the structure and properties of the substance.

### 3.2. Accumulation of OPEs in D. magna via Dietary Route

Different from the waterborne exposure, although the pre-exposed *S. obliquus* (contained 0.60 ± 0.07 and 2.04 ± 0.56 μg/g dw of TCEP, 3.34 ± 0.49 and 20.49 ± 4.11 μg/g dw of TDCPP, 0.52 ± 0.05 and 2.96 ± 0.39 μg/g dw of TBOEP, and 3.01 ± 0.60 and 16.69 ± 2.94 μg/g dw of TPHP) was continuously added, the internal concentration of *D. magna* reached a stable level within 3 h (Figure 2), and the level was much lower than the internal concentration in the waterborne exposure. Since the concentration of OPEs depurated from the algae into the solution was very low (lower than 1.5 and 6 μg/L under two exposure groups, respectively) (Appendix A), we ignored OPEs via waterborne exposure during the dietary exposure period. The depuration rate constant (*k_d_*_2_) of TCEP, TDCPP, TBOEP, and TPHP under two exposure concentrations were different from the values in waterborne exposure (Table 1). Another study also found that the depuration rate constant varied with different exposure routes [41]. This indicated that OPEs entering *D. magna* via different exposure routes might have different metabolic pathways, resulting in different depuration processes.

For the four OPEs, the exposure concentration changed from 20 μg/L to 100 μg/L, and the BMFs of TCEP, TDCPP, and TPHP increased from 0.35, 0.16, 0.04 to 0.61, 0.23, and 0.07, respectively (Table 1), indicating that the delivery efficiency increased as the concentration increased. The BMF of TBOEP decreases from 0.15 to 0.07, implying that as the concentration increases, the transfer efficiency decreases. In this study, we found analogous results in BCF. Only the BCF values of TBOEP decreased with an increasing exposure concentration. Therefore, TBOEP may not be easily accumulated in *D. magna* either via waterborne or dietary exposure. The possible reason was that TBOEP was more accumulative in algae and was quickly and easily transformed into bis(2-butoxyethyl) hydroxyethyl phosphate (BBOEHEP) and bis(2-butoxyethyl) 3-hydroxyl−2-butoxyethyl phosphate (3-OH-TBOEP) in *D. magna*, according to a previous study in salmon [37]. For TPHP, which could be biotransformed into diphenyl phosphate, hydroxylated triphenyl phosphate, and thiol triphenyl phosphate in *D. magna* [42], its transfer efficiency was also low (BMF = 0.045 and 0.074). TCEP and TDCPP may not be easily transformed in the algae, so they have a higher transmission efficiency. However, there are few studies on daphnia and algae. In future studies, it may be necessary to explore and compare the biotransformation ability of OPEs in algae and *D. magna* in order to fully explain the difference in transfer efficiency between the different types of OPEs.

In the field studies, there are no uniform conclusions on the biomagnification effect of OPEs, which may be related to the food web structure of the study area. For example, Bekele et al. reported the biomagnification of TBOEP, TEHP (tris(2-ethylhexyl) phosphate), TEP (triethyl phosphate), TCP (tricresyl phosphate), TCPP, and CDPP (cresyl diphenyl phosphate) with trophic magnification factors (TMFs) ranging from 1.75 to 2.72 [7], while the TMFs of TCEP, TDCPP, and TPHP were 5.20, 2.92, and 2.74, respectively [12]. TNBP and TPhP with TMF values of 0.72, 0.57, and 0.62, respectively, in a typical freshwater food web [5] and TMF values of TBP (tributyl phosphate), TCEP, TBOEP, and TEHP were all slightly higher than 1 in a food web of the Nansha Islands [43]. In our study, the BMFs of four OPEs were all below 1, which means trophic dilution in the transfer from *S. obliquus* to *D. magna*. To confirm the biomagnification effect of OPEs in further studies, it may be necessary to construct a more complex aquatic food chain or even a food web (a microcosm/mesocosm) for research. Furthermore, the possible impact of the food web structure may need to be taken into consideration.

### 3.3. Relative Importance of Waterborne and Dietary Routes to the Accumulation of OPEs in D. magna

In the algal exposure experiments, we performed a single exposure to *S. obliquus*. At 24 h, the internal concentration of almost all OPEs in *S. obliquus* reached the peak and then began to decrease (Appendix A). So we assumed that the internal concentration of algae at 24 h was steady under their exposure concentration. Then we calculated the contribution of OPEs uptake via the food route relative to the waterborne route (F) (Figure 3). On the whole, the value of F decreased with exposure time. This may be related to the bioaccumulation characteristics of waterborne and dietary exposure in *D. magna*. Within 3 h, the internal concentration of *D. magna* had reached a steady state via dietary exposure, while in waterborne exposure, the internal concentration increased within 12 h. For compounds with log Kow > 5, the uptake from food (dietary exposure), rather than from the dissolved phase (waterborne exposure), was a more important exposure route [33]. A previous study showed that BDE-47, with the log Kow value of 6.16, contributed more from food and was higher than that from filtration to bioaccumulation in *D. magna* [22]. In our study, except for TBOEP, the contribution of dietary exposure to the other three OPEs was lower than that of waterborne exposure. This may be because TBOEP had a stronger potential to accumulate in *S. obliquus* than in *D. magna*. Under the same exposure concentration, the internal concentration of *S. obliquus* at 24 h was higher than that of *D. magna*. It may also be because the biotransformation ability of TBOEP in *D. magna* was stronger than that in *S. obliquus*. Conversely, for the other three OPEs, especially for TPHP, it might be more bioaccumulative in *D. magna* (Figure 1) than in *S. obliquus* (Appendix A). On the other hand, the biotransformation ability of TPHP in *S. obliquus* was likely stronger than that in *D. magna*, so TPHP might be much easier to accumulate in *D. magna* via waterborne exposure.

Some other studies suggested that the differences between food conditions and between species need to be taken into consideration. Zhang et al. found that in the algae–copepods–fish food chain, dioxins via dietary exposure were more important for both copepods and fish, and it was more easily accumulated in the copepods with algae with moderate bioaccumulation ability for dioxins [41]. Wang and Wang reported that it was equally important for dichlorodiphenyltrichloroethane (DDT) to be accumulated via waterborne and dietary exposure in the copepods (*Acartia erythraea*), while waterborne exposure was a more important pathway than uptake from food in fish (*Lutjanus argentimaculatus*) [33].

In future research, it may be possible to focus on the transfer process of metabolites in the *S. obliquus*–*D. magna* food chain, in order to clarify the reasons for the differences in the contribution of two exposure routes in different OPEs. We may need to pay more attention to aquatic organisms at higher trophic levels such as mussels, fish, or frogs, because the contribution of dietary and waterborne exposure may be different from in *D. magna*. It might also be possible to trace the transformation process of OPEs with the isotope labeling method, such as in previous research dealing with bisphenol a [44] and BDE-47 [22]. At the same time, algae with different pollutant bioaccumulation abilities might be considered food for research.

## 4. Conclusions

The uptake and depuration kinetics of four OPEs via waterborne and dietary exposure has been investigated in *D. magna*. Results showed that in the waterborne experiment, TPHP has the highest uptake rate constant (*k_u_*_1_) and lowest depuration rate (*k_d_*_1_) constant, which means it is more bioaccumulative in *D. magna* than the other three OPEs. In comparison, TBOEP is the least bioaccumulative. In the dietary experiment, the depuration rate constant (*k_d_*_2_) was different from k_d1_, suggesting different metabolism and depuration processes in the two exposure routes. The BMFs of four OPEs were all below 1, indicating trophic dilution of OPEs in the aquatic food chain. Except for TBOEP, the contribution of dietary exposure to the other three OPEs was lower than that of waterborne exposure based on our hypotheses. Our study suggests that it is possible that, in the natural environment, OPEs are more easily accumulated by the waterborne exposure route in zooplankton. In future research, it might be necessary to pay attention to whether there are differences in the biotransformation of OPEs in organisms at different trophic levels. Meanwhile, the metabolic processes of OPEs in the food chain should receive more attention, as well as the possible differences in the biotransformation process in different exposure routes.

## Figures and Tables

**Figure 1 ijerph-19-09382-f001:**
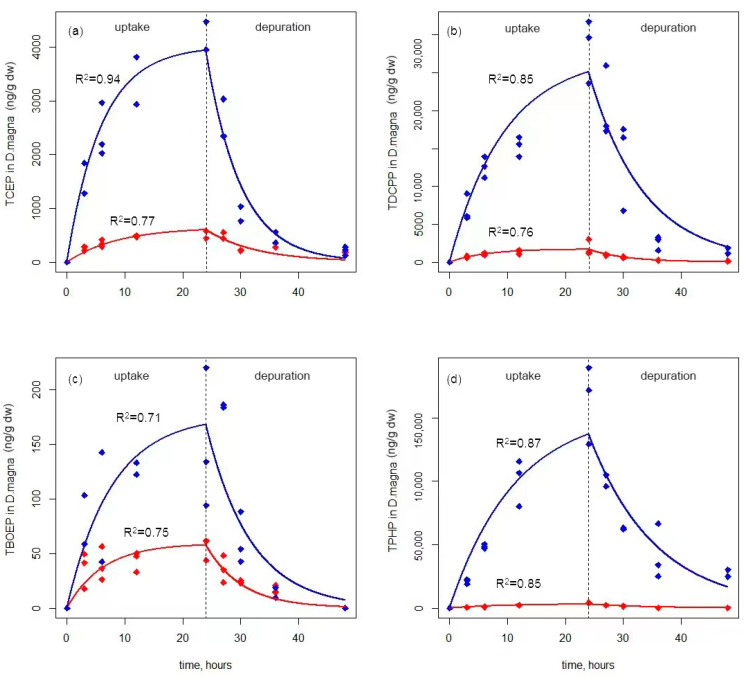
Uptake (0–24 h) and depuration (24–48 h) kinetics of four OPEs ((**a**) TCEP, (**b**) TDCPP, (**c**) TBOEP, (**d**) TPHP) in *D. magna* via waterborne exposure. Blue lines (toxicokinetics model simulations) and points (experimental data) represent the high-concentration group (100 μg/L), and red lines (toxicokinetics model simulations) and points (experimental data) represent the low-concentration group (20 μg/L).

**Figure 2 ijerph-19-09382-f002:**
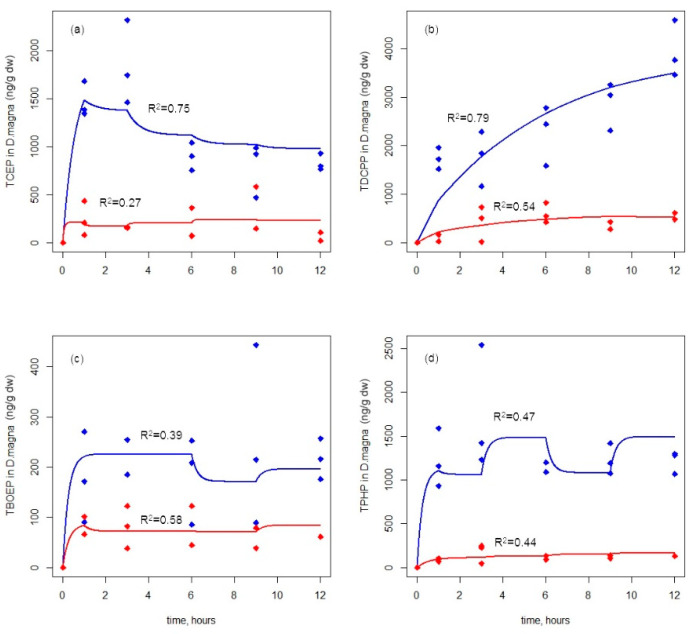
Uptake kinetics of four OPEs ((**a**) TCEP, (**b**) TDCPP, (**c**) TBOEP, (**d**) TPHP) in *D. magna* via dietary exposure. Blue lines (toxicokinetics model simulations) and points (experimental data) represent the high-concentration group (100 μg/L), and red lines (toxicokinetics model simulations) and points (experimental data) represent the low-concentration group (20 μg/L).

**Figure 3 ijerph-19-09382-f003:**
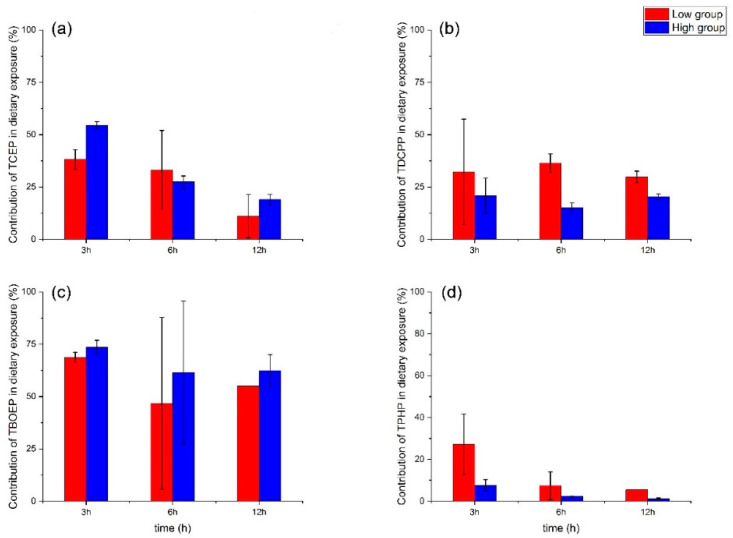
Contribution of OPEs ((**a**) TCEP, (**b**) TDCPP, (**c**) TBOEP, (**d**) TPHP) in *D. magna* via dietary exposure relative to waterborne exposure. Red bars represent the low-concentration group (20 μg/L), and blue bars represent the high-concentration group (100 μg/L). The error bar represents the standard error.

**Table 1 ijerph-19-09382-t001:** Parameters in uptake and depuration kinetics of OPEs in *D. magna*.

	Measured Exposure Concentration (μg/L)	Waterborne Exposure				Dietary Exposure		
		*k_u_*_1_ (L kg^−1^ h^−1^)	*k_d_*_1_ (h^−1^)	*t* _1/2_ (h)	BCF (L kg^−1^)	*k_u_*_2_ (h^−1^)	*k_d_*_2_ (h^−1^)	BMF
TCEP	21.72	3.39	0.12	6.03	29.48	5.82	16.41	0.35
	116.67	5.78	0.17	4.13	34.40	1.05	1.71	0.61
TDCPP	17.07	16.04	0.16	4.36	100.86	0.06	0.38	0.17
	124.22	23.22	0.11	6.60	221.14	0.03	0.15	0.22
TBOEP	16.71	0.57	0.16	4.28	3.52	0.53	3.59	0.15
	135.67	0.17	0.13	5.33	1.29	0.25	3.82	0.07
TPHP	16.29	21.15	0.10	6.66	203.37	0.09	2.05	0.04
	99.06	138.80	0.09	7.88	1577.27	0.29	3.88	0.07

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
