# Peer review of "Waterborne and Dietary Bioaccumulation of Organophosphate Esters in Zooplankton Daphnia magna"

_ijerph, 2022, doi:10.3390/ijerph19159382_

Round 1
Reviewer 1 Report
Review - ijerph-1829861
In the manuscript entitled ‘Waterborne and dietary bioaccumulation of organophosphate esters in zooplankton Daphnia magna’, the authors studied the waterborne and dietary routes leading to accumulation and depuration of several organophosphate pollutants using a well-conceived experimental laboratory protocol. Their experiment led to interesting results, notably on the different potentials of bioaccumulation and transfer within aquatic food-webs of these pollutants. They combined both experimental data and toxicokinetic modelling to better explore metabolic pathways and processes. On a general note, I found the methods and the results well-described. These results were compared with existing literature on other organisms, and implications of these results were for the most part interesting. Limitations and potential for future research were also described. Overall, the experimental work and the results described in this manuscript are of good quality. I believe that the main limitation of the study concerns the fact that the dietary route was investigated with only type of prey and one trophic level. In addition, I believe that some parts could be improved. I notably strongly suggest the use of an English correction service, as the quality of English in this manuscript must be improved. Similarly, a lot of small mistakes, mainly editorial, lowered the quality of the manuscript. I listed my specifics comments below.
Introduction
L36 – “it” should be replaced by “they”.
L38- “they easily diffuse into the surrounding environment” instead of “it is easy to diffuse into the surrounding environment”
L55- Maybe it could be a good idea to precise what type of organism Mugilogobius chulae is, at least that it is a fish.
L57- I think “Guo 2017b” should be “Guo et al. 2017b”. Also, I think that Cd should be spelled out when first written.
L58- NPs should also be spelled out I think.
L62. Same thing as my previous comment, I guess the reference is “de Wit et al. 2012” and not just “de Wit 2012”. Please make sure that references are correct in the entire manuscript.
L62-65. This sentence needs to be rewritten as for now it seems not grammatically correct for me.
L76. Even if D. magna is widely studied, I think that some explanations on what kind of organism it is should be added here.
L84. Same, a brief explanation of what kind of organism Scenedesmus obliquus is and why it is a good choice to study dietary exposure on D. magna. S. obliquus should also be put in italics, please correct here and elsewhere in the manuscript. On a more general note, please verify that all genus and species names are put in italics.
Materials and Methods
L104. I do not understand why reconstructed water was used instead of natural water. Please clarify.
L115. Please indicate filter pore size and diameter.
L132. The algal concentration must be correctly formatted.
L135-136. I do not understand this sentence, please reformulate and clarify.
Results
On a general note, this section presents both results and some discussion about these results, so maybe it could be renamed as ‘Results and Discussion’, but I leave this at the editor’s discretion.
L210-211. Why are there two values for the uptake rate constant for each OPE? Do these values correspond to duplicate experiments (if so, why not show the third triplicate)? Or do they represent the range of obtained results? I have the feeling that they represent the two tested concentrations. Please explain.
L237-239. This is a very interesting result, suggesting that the properties of the pollutants are more important than the properties of the organisms in the accumulation process. What could be the consequences and implications of such results?
L270-272. I think that another interesting point that should be explored in future studies is related to D. magna prey, maybe testing other preys than just S. obliquus could enable to elucidate more the dietary route.
L300-301. This sentence needs to be reformulated as it is unclear for me.
Conclusions
L357. I agree with this conclusion, however I think that it should be tempered by stating that this result comes from experiments in which dietary exposure was investigated with only one trophic level and one type of prey.
Author Response
Reviewer 1
In the manuscript entitled ‘Waterborne and dietary bioaccumulation of organophosphate esters in zooplankton Daphnia magna’, the authors studied the waterborne and dietary routes leading to accumulation and depuration of several organophosphate pollutants using a well-conceived experimental laboratory protocol. Their experiment led to interesting results, notably on the different potentials of bioaccumulation and transfer within aquatic food-webs of these pollutants. They combined both experimental data and toxicokinetic modelling to better explore metabolic pathways and processes. On a general note, I found the methods and the results well-described. These results were compared with existing literature on other organisms, and implications of these results were for the most part interesting. Limitations and potential for future research were also described. Overall, the experimental work and the results described in this manuscript are of good quality. I believe that the main limitation of the study concerns the fact that the dietary route was investigated with only type of prey and one trophic level. In addition, I believe that some parts could be improved. I notably strongly suggest the use of an English correction service, as the quality of English in this manuscript must be improved. Similarly, a lot of small mistakes, mainly editorial, lowered the quality of the manuscript. I listed my specifics comments below.
Thanks for your comments. We quite agree with your suggestion. Due to the limitation of experimental conditions, we only studied one predator and one trophic level. But we looked at the very important consumer of the ecosystem, zooplankton. They play an important role in connecting producers with higher-trophic consumers. Therefore, our results will lay the foundation for future research on more complex food chains.
Introduction
L36 – “it” should be replaced by “they”
Revised as suggested.
L38- “they easily diffuse into the surrounding environment” instead of “it is easy to diffuse into the surrounding environment”
Revised as suggested.
L55- Maybe it could be a good idea to precise what type of organism Mugilogobius chulae is, at least that it is a fish.
Revised as suggested.
L57- I think “Guo 2017b” should be “Guo et al. 2017b”. Also, I think that Cd should be spelled out when first written.
Revised as suggested.
L58- NPs should also be spelled out I think.
Revised as suggested.
L62. Same thing as my previous comment, I guess the reference is “de Wit et al. 2012” and not just “de Wit 2012”. Please make sure that references are correct in the entire manuscript.
Revised as suggested.
L62-65. This sentence needs to be rewritten as for now it seems not grammatically correct for me.
Revised as suggested.
L76. Even if D. magna is widely studied, I think that some explanations on what kind of organism it is should be added here.
Revised as suggested.
L84. Same, a brief explanation of what kind of organism Scenedesmus obliquus is and why it is a good choice to study dietary exposure on D. magna. S. obliquus should also be put in italics, please correct here and elsewhere in the manuscript. On a more general note, please verify that all genus and species names are put in italics.
Revised as suggested.
Materials and Methods
L104. I do not understand why reconstructed water was used instead of natural water. Please clarify.
As we mentioned in the introduction, OPEs can easily spread from products to natural environments. Therefore, OPEs have been detected in many major watersheds, oceans and lakes around the world with a certain concentration. In addition, the physical and chemical properties of water bodies in different regions are also different. Therefore, in order to avoid the influence of background concentration value and water quality conditions on the experimental results, we used reconstructed water instead of natural water.
L115. Please indicate filter pore size and diameter.
Revised as suggested.
L132. The algal concentration must be correctly formatted.
Revised as suggested.
L135-136. I do not understand this sentence, please reformulate and clarify.
Revised as suggested.
Results
On a general note, this section presents both results and some discussion about these results, so maybe it could be renamed as ‘Results and Discussion’, but I leave this at the editor’s discretion.
L210-211. Why are there two values for the uptake rate constant for each OPE? Do these values correspond to duplicate experiments (if so, why not show the third triplicate)? Or do they represent the range of obtained results? I have the feeling that they represent the two tested concentrations. Please explain.
The two values for the uptake rate constant represent the value in two tested concentrations, respectively. They come from the simulation results of three parallel values in two experimental groups, respectively.
L237-239. This is a very interesting result, suggesting that the properties of the pollutants are more important than the properties of the organisms in the accumulation process. What could be the consequences and implications of such results?
Structures of OPEs vary depending on different ester linkages. OPEs can be roughly divided into three types according to their different structures: chlorinated OPEs, alkyl-OPEs and aryl-OPEs[1]. OPEs have a wide range of physiological properties in the environment. These properties are important factors for assessing influence of OPEs on organisms[2]. In our study, we select TCEP and TDCPP to represent chlorinated OPEs, TBOEP to represent alkyl-OPEs and TPHP to represent aryl-OPEs. So our results may suggest that it is important to pay more attention to the impact of the structure of OPEs.
L270-272. I think that another interesting point that should be explored in future studies is related to D. magna prey, maybe testing other preys than just S. obliquus could enable to elucidate more the dietary route.
Thanks for your comment. We totally agree with you. In future research, other preys such as Chlorella pyrenoidosa, Chlamydomonas reinhardtii and Cyanobacteria might be taken into consideration.
L300-301. This sentence needs to be reformulated as it is unclear for me.
Revised as suggested.
Conclusions
L357. I agree with this conclusion, however I think that it should be tempered by stating that this result comes from experiments in which dietary exposure was investigated with only one trophic level and one type of prey.
Revised as suggested.
Reviewer 2 Report
On account of the manuscript IJERPH-1829861, entitled “Waterborne and dietary bioaccumulation of organophosphate esters in Zooplankton Daphnia magna” by Wenxin Liu et al., the authors evaluated the bioaccumulation of 4 typical organophosphate esters (OPEs) (tris(2-chloroethyl) phosphate (TCEP), tris(1,3-dichloro-2-propyl) phosphate (TDCIPP), tris(2-butoxyethyl) phosphate (TBOEP), and triphenyl phosphate (TPHP)) and their toxicokinetics under waterborne and dietary exposure routes. The topic is important to conduct environmental risk assessment of OPEs in the aquatic environment. After careful consideration, I feel that this manuscript is to be published after improvement of some minor shortcomings. Details of my comments are as follows:
The manuscript was well written and designed, and the authors got interesting results. Several important revisions are, however, required before publication. The authors evaluated the ecotoxicological effects in two systems, one with high (100 μg/L) and one with low (20 μg/L) exposure level to OPEs. However, discussion of statistically significant differences between the results of the two groups, and changes in toxic effects during exposure experiment, was not included in the present manuscript, which would critically affect the accuracy and reliability of the results. The authors are better to include these aspects in more detail, and deepen the results and discussions for the issues mentioned above. In addition, Statistical Analysis is also needs to be included in 2. Materials and Methods Section. The authors are strongly encouraged to take these aspects into account to show the results for enhancement of the accuracy and better understanding of the results. After that I am ready to recommend the present manuscript for publication.
Author Response
Reviewer 2
On account of the manuscript IJERPH-1829861, entitled “Waterborne and dietary bioaccumulation of organophosphate esters in Zooplankton Daphnia magna” by Wenxin Liu et al., the authors evaluated the bioaccumulation of 4 typical organophosphate esters (OPEs) (tris(2-chloroethyl) phosphate (TCEP), tris(1,3-dichloro-2-propyl) phosphate (TDCIPP), tris(2-butoxyethyl) phosphate (TBOEP), and triphenyl phosphate (TPHP)) and their toxicokinetics under waterborne and dietary exposure routes. The topic is important to conduct environmental risk assessment of OPEs in the aquatic environment. After careful consideration, I feel that this manuscript is to be published after improvement of some minor shortcomings. Details of my comments are as follows:
The manuscript was well written and designed, and the authors got interesting results. Several important revisions are, however, required before publication. The authors evaluated the ecotoxicological effects in two systems, one with high (100 μg/L) and one with low (20 μg/L) exposure level to OPEs. However, discussion of statistically significant differences between the results of the two groups, and changes in toxic effects during exposure experiment, was not included in the present manuscript, which would critically affect the accuracy and reliability of the results. The authors are better to include these aspects in more detail, and deepen the results and discussions for the issues mentioned above. In addition, Statistical Analysis is also needs to be included in 2. Materials and Methods Section. The authors are strongly encouraged to take these aspects into account to show the results for enhancement of the accuracy and better understanding of the results. After that I am ready to recommend the present manuscript for publication.
Thanks for your comment. We took a minor revision as suggested. In our study, although we construct two concentration groups, we focused on comparing the differences of the 4 OPEs in different exposure routes within each concentration group, rather than the differences between concentration groups. Therefore, we did not perform statistical analyses between concentration groups. We set up three parallel experiments at each time point in the two concentration groups of the two exposure routes. In the model simulations, the ODE method was used to optimize, and three sets of test values were fitted simultaneously to obtain the final optimization result (Fig. 1 and Fig2). In Figure 3, we use error bars to represent standard deviations.
Reviewer 3 Report
The research is interesting and actual. It tries to bring more knowlegde about the process. The study is well designed and the methods are adequate.
More detail ajustment curves done with data shown in figure 1 and 2 could be very helpful and informative.
A long the text the way the numbers (powers of ten and index) and units are typed is not correct ( ex. lines 109 and 132, 136, etc.. The same with the number in chemical formulae (ex. lines 104, 105 and 161).
The types of the variables in the text are not in agreement with the types used in the equations (ex. line 179.
It is not clear of the meaning of the abrreviation dw that appears in line 192 and figure 1 (destiled water?).
I signed in the pdf thing that should be corrected (yellow, and green) reviewed ( blue and pink) or delelet (in red).
In the same way Daphina magnus is abbreviated as D. magna Scenedesmus obliquus should be S. obliquus.
The latin expression et al. also should be typed as et al.
The titles of the figures should be edited in order to me in accordance with the test types and form. Attention to the spaces in value/unit in the legend of the figures.
Finally importante warning with the significant figures when expressing quantative values (as in line 165, 243, 244 and Table 1). In line 165: 73.34 %+/-2.06 % shoul be just 73 +/- 2 %. In line we shold have 595 +/- 73 ng/g
and 2041 +/- 558 shoul be 2,04 +/- 0,56 microgram /gram!

Author Response
Reviewer 3
The research is interesting and actual. It tries to bring more knowlegde about the process. The study is well designed and the methods are adequate.
More detail ajustment curves done with data shown in figure 1 and 2 could be very helpful and informative.
Revised as suggested.
A long the text the way the numbers (powers of ten and index) and units are typed is not correct ( ex. lines 109 and 132, 136, etc.. The same with the number in chemical formulae (ex. lines 104, 105 and 161).
Revised as suggested.
The types of the variables in the text are not in agreement with the types used in the equations (ex. line 179.
Revised as suggested.
It is not clear of the meaning of the abrreviation dw that appears in line 192 and figure 1 (destiled water?).
Revised as suggested (Line118, line132 and line148).
I signed in the pdf thing that should be corrected (yellow, and green) reviewed ( blue and pink) or delelet (in red).
Corrected in the revised ms.
In the same way Daphina magnus is abbreviated as D. magna Scenedesmus obliquus should be S. obliquus.
Revised as suggested.
The latin expression et al. also should be typed as et al.
Revised as suggested.
The titles of the figures should be edited in order to me in accordance with the test types and form. Attention to the spaces in value/unit in the legend of the figures.
In Figures 1 and 2, the vertical axis represents the concentration of OPEs in the D. magna and the horizontal axis represents time. So we simplify the name of vertical axis as “OPEs in D. magna”. The spaces in value/unit have been revised as suggested.
Finally importante warning with the significant figures when expressing quantative values (as in line 165, 243, 244 and Table 1). In line 165: 73.34 %+/-2.06 % shoul be just 73 +/- 2 %. In line we shold have 595 +/- 73 ng/g and 2041 +/- 558 shoul be 2,04 +/- 0,56 microgram /gram!
Revised as suggested. But we keep two decimal places in table 1.
References:
- Hou, R.; Xu, Y.; Wang, Z., Review of OPFRs in animals and humans: Absorption, bioaccumulation, metabolism, and internal exposure research. Chemosphere 2016, 153, 78-90.
- van der Veen, I.; de Boer, J., Phosphorus flame retardants: properties, production, environmental occurrence, toxicity and analysis. Chemosphere 2012, 88, (10), 1119-53.